# Biocompatible Magnetic Colloidal Suspension Used as a Tool for Localized Hyperthermia in Human Breast Adenocarcinoma Cells: Physicochemical Analysis and Complex In Vitro Biological Profile

**DOI:** 10.3390/nano11051189

**Published:** 2021-04-30

**Authors:** Elena-Alina Moacă, Claudia-Geanina Watz, Vlad Socoliuc, Roxana Racoviceanu, Cornelia Păcurariu, Robert Ianoş, Simona Cîntă-Pînzaru, Lucian Barbu Tudoran, Fran Nekvapil, Stela Iurciuc, Codruța Șoica, Cristina-Adriana Dehelean

**Affiliations:** 1Faculty of Pharmacy, Victor Babeș University of Medicine and Pharmacy Timisoara, 2nd Eftimie Murgu Square, RO-300041 Timisoara, Romania; alina.moaca@umft.ro (E.-A.M.); farcas.claudia@umft.ro (C.-G.W.); codrutasoica@umft.ro (C.Ș.); cadehelean@umft.ro (C.-A.D.); 2Research Centre for Pharmaco-Toxicological Evaluation, “Victor Babeș” University of Medicine and Pharmacy, 2nd Eftimie Murgu Square, RO-300041 Timișoara, Romania; 3Romanian Academy—Timisoara Branch, Center for Fundamental and Advanced Technical Research, Laboratory of Magnetic Fluids, 24 M. Viteazu Ave., RO-300223 Timisoara, Romania; 4Research Center for Complex Fluids Systems Engineering, Politehnica University of Timisoara, 1 M. Viteazu Ave., RO-300222 Timisoara, Romania; 5Faculty of Industrial Chemistry and Environmental Engineering, Politehnica University of Timisoara, 2nd Victoriei Square, RO-300223 Timisoara, Romania; cornelia.pacurariu@upt.ro (C.P.); robert_ianos@yahoo.com (R.I.); 6Biomolecular Physics Department, Babes-Bolyai University, 1 Kogalniceanu Street, RO-400084 Cluj-Napoca, Romania; simona.pinzaru@ubbcluj.ro (S.C.-P.); neki.fran@gmail.com (F.N.); 7RDI Laboratory of Applied Raman Spectroscopy, RDI Institute of Applied Natural Sciences (IRDI-ANS), Babeş-Bolyai University, 42 Fântânele Street, RO-400293 Cluj-Napoca, Romania; 8Electron Microscopy Laboratory “Prof. C. Craciun”, Faculty of Biology & Geology, “Babes-Bolyai” University, 5-7 Clinicilor Street, RO-400006 Cluj-Napoca, Romania; lucianbarbu@yahoo.com; 9Electron Microscopy Integrated Laboratory, National Institute for R & D of Isotopic and Molecular Technologies, 67-103 Donat Street, RO-400293 Cluj-Napoca, Romania; 10Department of Cardiology, Faculty of Medicine, Victor Babes University of Medicine and Pharmacy, 2nd Eftimie Murgu Square, RO-300041 Timisoara, Romania; siurciuc@umft.ro

**Keywords:** magnetic iron oxides nanoparticles, citric acid, Raman spectroscopy, combustion method, breast adenocarcinoma

## Abstract

Magnetic iron oxide nanoparticles are the most desired nanomaterials for biomedical applications due to their unique physiochemical properties. A facile single-step process for the preparation of a highly stable and biocompatible magnetic colloidal suspension based on citric-acid-coated magnetic iron oxide nanoparticles used as an effective heating source for the hyperthermia treatment of cancer cells is presented. The physicochemical analysis revealed that the magnetic colloidal suspension had a z-average diameter of 72.7 nm at 25 °C with a polydispersity index of 0.179 and a zeta potential of −45.0 mV, superparamagnetic features, and a heating capacity that was quantified by an intrinsic loss power analysis. Raman spectroscopy showed the presence of magnetite and confirmed the presence of citric acid on the surfaces of the magnetic iron oxide nanoparticles. The biological results showed that breast adenocarcinoma cells (MDA-MB-231) were significantly affected after exposure to the magnetic colloidal suspension with a concentration of 30 µg/mL 24 h post-treatment under hyperthermic conditions, while the nontumorigenic (MCF-10A) cells exhibited a viability above 90% under the same thermal setup. Thus, the biological data obtained in the present study clearly endorse the need for further investigations to establish the clinical biological potential of synthesized magnetic colloidal suspension for magnetically triggered hyperthermia.

## 1. Introduction

Magnetic iron oxide nanoparticles, termed MIONPs, have become one of the most desirable nanomaterials due to their unique physiochemical properties (e.g., stability in solutions at neutral pHs) [1,2], high biocompatibility and biodegradability [3], low toxicity as compared with other metallic nanoparticles, long-time retention, superparamagnetic properties [4,5,6,7], and surface modulation by chemical functionalization [8]. Magnetite (Fe_3_O_4_) and its oxidized form maghemite (γ-Fe_2_O_3_) are two important iron oxides that share the characteristics listed above, thus being excellent candidates for biomedical applications and in vitro/in vivo studies. Various methods regarding the fabrication of MIONPs have been reported that are not harmful to the environment and show promisingly high compatibility with human physiology. From all the synthesis methods, one that has caught the attention of researchers is the combustion method, a synthesis method brought to light approximatively 35 years ago by Patil [9]. The huge benefit of this method is its reliability in the synthesis of different oxide materials, alloys, magnetic materials, and composites [10,11,12,13]. This synthesis method presents many advantages compared to other synthesis methods. Among them are that it requires low amounts of energy because it consists of a single-step process without further annealing at high temperatures, has a short duration (in the range of minutes), and is environmentally friendly.

In order to prevent the agglomeration of magnetic nanoparticles (formation of large aggregates and biodegradation following exposure to a biological system) and to ensure high biocompatibility, the MIONPs are usually coated with an organic species (surfactants, polymers), inorganic layers (silica, gold), or small molecules during or after the synthesis process [8,14,15,16,17,18,19]. The coating acts as shield covering the magnetic nanoparticles’ surface, which can also be functionalized by attaching other molecules (through covalent attachment, adsorption, or entrapment on the nanoparticles) in order to increase the targeting yield [20,21]. Electrostatic stabilization compared with the polymeric one presents the advantage of using small molecules with a large deprotonating ability, which insignificantly modifies the MIONPs’ hydrodynamic behavior and is of great importance in magnetic hyperthermia approaches for potential clinical applications.

Small molecules (e.g., citric acid molecules) are quite attractive due to their simple chemical conjugation (chemisorption) onto the magnetic iron oxide surface by binding to the Fe-OH molecules, thus leading to the formation of carboxylate groups. Citric acid (CA) has three carboxyl groups and one hydroxyl group that provide stability and negatively high surface area, making them suitable for further functionalization [22]. For medical diagnoses and therapies, it is necessary for MIONPs to be stable in hydrophilic mediums at a neutral pH and physiological salinity. Colloidal stability in aqueous and physiological mediums represents one of the most important issues related to the biomedical applications of magnetic nanoparticles. In this regard, one of the key factors to achieve colloidal stability is the dimension of the magnetic nanoparticles, which should possess an adequate diameter to avoid sedimentation due to gravitational forces (<100 nm) [23,24]. Thus, the nanoparticles are capable of trespassing through the altered capillary systems of cancerous organs and tissues by taking advantage of the enhanced permeability and retention (EPR) effect, avoiding vessel embolism.

Highly stable and biocompatible colloidal suspensions based on magnetic nanoparticles (ferrofluids) are extensively investigated for magnetic fluid hyperthermia therapies (MFHT) in cancer treatment. Hyperthermia is deemed a noninvasive technique for cancer therapy due to the fact that kills activated cancerous cells via localized heating of tumors at supranormal body temperatures (40–46 °C). The most adequate nanomaterials used for MFHT cancer treatment are magnetic nanoparticles (Fe_3_O_4_ and γ-Fe_2_O_3_) [25], which must be relatively monodispersed, highly crystalline, water dispersible, biocompatible with high magnetic susceptibility, and display superparamagnetic behavior [26]. Superparamagnetic iron oxide nanoparticles have been widely used as hyperthermia agents mainly due to their strong magnetic properties and nanosized diameter (range of 20–100 nm), which can ensure an intense heating by alternating the magnetic fields applied [27]. Recently, Reyes-Ortega and coworkers [28], synthesized spherical, cuboidal, or rod-like shaped magnetic particles in the 40 nm size range using the hydrothermal synthesis route with the idea that these three types of magnetic nanoparticles could be used as hyperthermia agents. Using a magnetic field of 20 kA/m amplitude, they found that the cuboidal magnetic nanoparticles displayed a maximum ILP value of 3.0 ± 0.2 nHm^2^/kg, which is the highest value recorded for cubic magnetite nanoparticles. Hence, the authors showed that cubic magnetite nanoparticles are very attractive as a potent hyperthermia agent that can be applied at low, biomedically safe frequencies [28]. In another recent study conducted by Rivera-Chaverra and coworkers [29], the authors synthesized iron oxide nanoparticles using the laser ablation technique, varying the power of the laser, for possible applications in magnetic hyperthermia. They found that the heating capacity of magnetic nanoparticles tends to decrease with increasing laser energy.

There are three types of hyperthermia reported by the National Cancer Institute (www.nci.nih.gov, accessed on 24 November 2020): local hyperthermia (in which a small area of the tumor tissue is heated), regional hyperthermia (in which large areas of tissue and the whole body are heated), and whole-body hyperthermia. For these hyperthermia treatments, the MIONPs are spread throughout the target tissue via direct injection and then the tissue is exposed to an alternate magnetic field (AMF) that has the property of heat generation by one of the two magnetic relaxation mechanisms [30]. Salimi and collaborators [31] applied magnetic hyperthermia as a treatment method in breast cancer using dendrimers formed from a functionalized iron oxide nanoparticle with poly amidoamine. The experiment revealed an increased cellular apoptosis in cancer cells and a decrease of the tumor mammary gland growth. Brero and coworkers [32] proposed a combined therapy based on radiotherapy and hyperthermia to be used for the treatment of pancreatic cancer (BxPC3 cells). The experimental results obtained ensured the smooth transposition of the proposed protocol to the clinic.

The present work reports a facile single-step process for preparation of a highly stable and biocompatible magnetic colloidal suspension (MCS) based on citric-acid-coated magnetic iron oxide nanoparticles to be used as an effective heating source for the hyperthermia treatment of cancer cells.

The main aim of the current study was the synthesis of magnetic iron oxide nanoparticles (MIONPs) with excellent magnetic properties by employing the combustion method, and further to obtain an MCS based on these MIONPs that manifest high stability in a hydrophilic medium under both normothermic and hyperthermic conditions (date DLS 37/43 °C). The stability of the MIONPs in an aqueous solution is based on creating electrostatic repulsion (due to ionic compounds) between nanoparticles that can overcome the nanoparticles’ magnetic interaction.

The second aim was to evaluate the hyperthermic efficacy of the as-synthesized MCS based on MIONPs. For this purpose, we used two genotypically distinctive breast adenocarcinoma cell lines (MDA-MB-231 and MCF-7) and one nontumorigenic cell line (MCF-10) under standard and hyperthermic conditions induced by means of an incubator, as previously described by Kossatz et al. [33,34], using the cumulative equivalent minutes protocol of 43 °C (CEM43).

## 2. Materials and Methods

### 2.1. Materials and Chemicals

For the synthesis of both MIONPs and MCS, the following reagents were used: iron nitrate nonahydrate (Fe(NO_3_)_3_·9H_2_O, Carl Roth, ≥96% pure, Karlsruhe, Germany), citric acid monohydrate (C_6_H_8_O_7_·H_2_O, ≥99.5% purity, Silal Trading, Bucuresti, Romania), distilled water (Chemical Company SA, Iasi, Romania), ethanol 96% *v/v* (C_2_H_5_-OH, Chemical Company SA, Iasi, Romania), and ammonia 25% (NH_3_, Chemical Company SA, Iasi, Romania). All reagents were used as received.

### 2.2. Cell Culture Media and Cell Lines

The cell culture media used were: high glucose Dulbecco’s Modified Eagle’s Medium (DMEM), Eagle’s Minimum Essential Medium (EMEM), and Dulbecco’s Modified Eagle’s Medium and Ham’s F12 medium (1:1 mixture; DMEM:F12), acquired from American Type Culture Collection (ATCC, Manassas, VA, USA). Of the cell lines tested in this study, two genotypically distinctive breast adenocarcinoma cell lines (MDA-MB-231—ATCC^®^ HTB-26 TM, MCF-7—ATCC^®^ HTB-22 TM) and one nontumorigenic cell line (MCF-10—ATCC^®^ CRL-10317) were obtained from ATCC (Manassas, VA, USA) as frozen items.

### 2.3. Experimental Methodology

#### Synthesis of MCS Based on MIONPs

MIONPs were prepared using the combustion method thoroughly described by Ianos et al., 2012 [35]. Briefly, 0.09 mol of Fe(NO_3_)_3_·9H_2_O and 0.15 mol of C_6_H_8_O_7_·H_2_O were dissolved in 25 mL of distilled water and rapidly heated in a round bottom flask in the absence of air using a heating mantle at 400 °C for several minutes. After the combustion reaction was finished, the MIONPs were hand-grinded and washed three times with distilled water. In the washing process, the MIONPs were magnetically separated from the supernatant using a neodymium magnet and after that dried at 70 °C in an oven. The recipe was design in order to obtain 0.03 mol of Fe_3_O_4_ and with an excess of fuel (double the stoichiometric amount) but applying a purity correction to Fe(NO_3_)_3_·9H_2_O, 98%), according to Equation (1):54Fe(NO_3_)_3_ + 46C_6_H_8_O_7_ ⇔ 18Fe_3_O_4_ + 276CO_2_ + 184H_2_O + 81N_2_(1)

Employing a slightly modified version of the method used by Wang et al. [36], the MIONPs were coated with citric acid in order to obtain the magnetic colloidal suspension. The protocol was as following: 1000 mg of MIONPs were dispersed by sonication (2 h at 50% amplitude using a QSonica Ultrasonic Liquid Processor 700W, Q700 Sonicator (Newtown, CT, USA)) in a glass flask containing 210 mL of a water/ethanol 20:1 solution after it had been soaked for 2 days. When the mixture temperature reached 80–82 °C, under vigorous stirring, the desired amount of citric acid (1 g/mL in distilled water) was added dropwise into the suspension and the reaction was kept at 80 °C for 30 min. After cooling at room temperature (24 °C), the solid obtained was magnetically separated and the MIONPs were washed three times with warm distilled water (60–70 °C) until all soluble impurities were removed. The pH of the MCS was adjusted with a 25% NH_3_ solution until a basic value was reached. Finally, the magnetic iron oxide nanoparticle monolayers were coated with a surfactant (MIONPs@CA) and dispersed in distilled water, filtered through a dense paper filter (d = 125 mm, Prat Dumas, Couze-et-Saint-Front, France) and then by a 33 mm diameter sterile syringe filter with a 0.22 µm pore size hydrophilic polyethersulfone (PES) membrane (Millex-GP, Merck, Darmstadt, Germany). Thus, we were able to obtain the stable and biocompatible magnetic colloidal suspension (MCS).

### 2.4. Characterization Technique of Naked MIONPs

The resulted naked MIONPs were structurally and magnetically characterized in terms of phase composition by X-ray powder diffraction (XRD), thermal behavior (thermogravimetry (TG) and differential scanning calorimetry (DSC)), magnetic properties, Raman spectroscopy, and scanning electron microscopy (SEM).

#### 2.4.1. Powder X-ray Diffraction (XRD)

X-ray diffraction (XRD) was employed to determine the crystal structure and the phase composition of naked MIONPs using a Rigaku ULTIMA IV diffractometer (Rigaku, Tokyo, Japan) using monochromated Cu-K_α_ radiation (λ = 0.15406 nm) at room temperature (24 °C) in the range of 10–80° in the 2θ scale operating at 40 kV and 40 mA. For peak assignment, the following PDF files were used: 1,011,267 (hematite, α-Fe_2_O_3_), 1,011,032 (magnetite, Fe_3_O_4_), and 9,006,318 (maghemite, γ-Fe_2_O_3_) from the International Centre for Diffraction Data Powder Diffraction File (ICDD PDF) 4+ 2019 data. The average crystallite size was calculated based on the XRD patters using the Debye–Sherrer equation:(2)$$D_{XRD}=\frac{0.9\times \lambda }{\beta \times \cos\theta }$$
where *D_XRD_* represented the crystallite size (nm), *λ* the radiation wavelength (nm), *β* the full width at half of the maximum (radians), and *θ* the Bragg angle.

#### 2.4.2. Thermal Behavior of Naked MIONPs

Thermogravimetry–differential scanning calorimetry (TG–DSC), an analytical technique, was used to assess the stability and composition of organic compounds contained in the naked MIONPs. Thermal behavior was studied in the temperature range of 10–1000 °C using a Netzsch STA 449C (Netzsch Co, Selb/Bavaria, Germany) instrument equipped with alumina crucibles. The TG and DSC curves were recorded at a heating rate of 10 K min^−1^ under a dynamic air atmosphere at a flow rate of 20 mL min^−1^.

#### 2.4.3. Magnetic Measurements

The magnetic properties of nanoparticles were determined via vibrating sample magnetometry (VSM) at room temperature (24 °C), using a VSM 880 ADE/DMS magnetometer (DMS/ADE Technologies, Westwood, MA, USA).

The intrinsic loss power (ILP) of the MCS in HF AC magnetic fields was measured using in-house built equipment. A commercial HF AC magnetic field generator was used with a 100 kHz frequency and 40–130 Oe field amplitude range. The sample temperature was measured using an OPTOCON-FOTOEMP1 (Optocon AG, Dresden, Germany) thermometer with a TS3 fiber optic sensor.

#### 2.4.4. Raman Spectroscopy of Naked MIONPs

Raman spectroscopy analysis was conducted using multiple instruments and excitation wavelengths due to the inherent low scattering properties of the samples. FT-Raman spectra were recorded using an Equinox 55 Bruker FT-IR spectrometer (Karlsruhe, Germany) with an integrated FRA-106S Raman module. A nitrogen-cooled Ge detector and a 1064 nm line excitation from an Nd: YAG laser was employed. A Renishaw InVia Reflex confocal Raman microscope, using a 632.8 nm He-Ne excitation and 100× objective (NA 0.9) provided the best output in terms of signal-to-background ratio. For measurement optimization, sampling droplets of colloidal nanoparticle suspension were drop-coated on a SpectRim TM hydrophobic microscopy plate. An excitation line from a Cobalt diode pumped solid state laser at 532 nm, an He-Ne laser operating at 632.8 nm, and a laser diode at 785 nm were employed for excitation. Parameter optimization was controlled from the Wire 3.4 software of the instruments. The spectral resolution was 0.5 cm^−1^ for visible line excitation and 1 cm^−1^ for NIR.

#### 2.4.5. SEM Analysis of Naked MIONPs

Morphology, qualitative, and semiquantitative analysis of naked MIONPs was carried out using the scanning electron microscope Hitachi SU8230 cold field emission gun STEM (Chiyoda, Tokyo, Japan) with EDX detectors X-Max^N^ 80 from Oxford Instruments (UK). The magnetic nanoparticles were sputter-coated with 6 nm of gold (Agar Automatic Sputtercoater, UK) for better conductivity, which is required for high resolution SEM imaging. SEM analysis parameters were HV (high vacuum) mode, 30 kV acceleration voltage, secondary electron detectors (upper and lower), and one of two magnification orders, one for a general aspect of samples and a higher one for surface topography analysis. The identified chemical species were expressed in atomic relative percent (At%).

### 2.5. Characterization Technique of MCS

The resulted MCS, based on magnetic iron oxide nanoparticle monolayers coated with CA, was characterized by Raman Spectroscopy, VSM and ILP, transmission electron microscopy (TEM), and dynamic light scattering (DLS).

#### 2.5.1. TEM Analysis of MCS

The particle size and morphology of the magnetic colloidal suspension, obtained by coating the MIONP monolayers with CA and dispersing them in distilled water, were determined by transmission electron microscopy (TEM) using a Hitachi HD2700 cold field emission gun STEM (Chiyoda, Tokyo, Japan) equipped with two windowless EDX detectors (X-Max^N^ 100). To prepare a TEM sample, a drop of magnetic colloidal suspension (7 µL) was placed on a carbon-coated copper grid and dried at room temperature (24 °C). The micrographs were obtained at 200 kV acceleration voltage.

#### 2.5.2. Hydrodynamic Diameter and Zeta Potential Measurements of MCS

The hydrodynamic diameter (H_d_), polydispersity index (PDI), and zeta potential were measured via dynamic light scattering using a Zatasizer Nano ZS from Malvern Instruments (Worcestershire, UK). Through the use of photon correlation spectroscopy, the particle size of MIONPs@CA were measured in aqueous suspension in a range from 0.4 nm to 9 µm. Zeta potential was measured by an electrophoretic light scattering technique using a flow cell. The measurement condition was: temperature 25 °C distilled water (as a dispersant medium) with a refractive index of 1.3328 and viscosity 0.8878 cP. Due to the fact that the MCS was designed in order to evaluate its hyperthermic efficacy within a biological environment, the sample was also analyzed at 37 °C and 43 °C.

### 2.6. Cell Culture Procedure

MDA-MB-231 cell lines were cultured in high glucose DMEM media supplemented with 10% fetal calf serum (FCS), MCF7 cell lines were grown in EMEM media containing 15% FCS, and nontumorigenic MCF-10A cell lines were cultured in DMEM:F12 media supplemented with 20 ng/mL epidermal growth factor (EGF), 0.01 mg/mL insulin, 500 ng/mL hydrocortisone, and 5% FCS. All the cell culture media were supplemented with an antibiotic mixture of 100 U/mL penicillin and 100 g/mL streptomycin to avoid microbial contamination. Eppendorf T75 flasks were used for the cell culture of each cell line and the cells were passaged every time the confluence was above 90% by using trypsin-EDTA. In addition, the specific cell culture media were renewed every two days. The cells were cultured under standard conditions: humidified atmosphere with 5% CO_2_ and 37 °C using a Steri-Cycle i160 incubator (Thermo Fisher Scientific, Inc., Waltham, MA, USA).

### 2.7. Thermal Setup Protocol

To induce the hyperthermic conditions necessary to evaluate the impact of the high-temperature treatment on cell behavior, the cumulative equivalent minutes method was employed at 43 °C, as described by Kossatz and collaborators [33,34]. In brief, the cells treated with MCS at concentrations of 6 and 30 µg/mL were maintained in an incubator for 90 min at 43 °C and afterward the temperature was modified to 37 °C to provide standard conditions of cell culturing over a period of 24 h.

### 2.8. Cell Viability Assessment

The cytotoxic effect was assessed on two different adenocarcinoma cell lines (MDA-MB-231 and MCF-7 cells) and one nontumorigenic cell line (MCF-10A cells) using 3-(4,5-dimethylthiazol-2-yl)-2,5-diphenyltetrazolium bromide in MTT proliferation tests. 200 µL cell culture media containing 1 × 10^4^ cells/well were seeded onto 96-well plates for 24 h. On the following day, each medium was replaced with 100 µL of fresh medium containing the test concentrations of MCS (6 and 30 µg/mL) and incubated for 24 h. The next day, 10 μL MTT reagent was added to each well before a three-hour incubation period at 37 °C. The last step consisted of adding 100 μL of a solubilization buffer to each well and incubation for 30 min at room temperature in a dark chamber. The absorbance of the samples was spectrophotometrically determined using a microplate reader (xMark™Microplate, Biorad, Bio-Rad Laboratories, Hercules, CA, USA) at 570 nm test wavelength.

Cell viability rate was quantified by employing the following formula:(3)$$Cell\;viability\;\left(\%\right)=\frac{Abs_{MCS\;}\pm \;HT_{\left(with\;cells\right)\;}-Abs_{MCS\;}\pm \;HT_{\left(without\;cells\right)}}{Abs_{CM}\;\pm \;HT_{\left(with\;cells\right)}-Abs_{CM}\;\pm \;HT_{\left(without\;cells\right)}}\times 10$$

Legend: *Abs_MCS_*, absorbance value of the cell media containing *MCS*

*Abs_CM_*, absorbance value of the cell media

*HT*, hyperthermia

### 2.9. Cellular Localization of MCS by Means of Prussian Blue Staining

A number of 5 × 10^5^ cells/well were seeded in six-well plates) and incubated to reach an optimal confluence. All the cell lines (MDA-MB-231, MCF-7, and MF-10A) were stimulated for 24 h with MCS at concentrations of 6 and 30 µg/mL. The experiments were conducted in standard and hyperthermic conditions, applying a slightly altered version of the protocols employed by Jadhav and collaborators [37]. In brief, after the incubation period passed (24 h), the cells were washed twice with 1.5 mL PBS/well and fixed with 4% paraformaldehyde (PFA) at 4 °C for 30 min. Thereafter, the cells were stained with an equal volume of freshly prepared 5% potassium ferrocyanide trihydrate and 5% hydrochloric acid in PBS for 30 min. The cells were washed again with PBS, counterstained with 1% neutral red for 5 min, and further distained by washing the cells three times with PBS. The cells were observed with an Olympus IX73 microscope (Olympus, Tokyo, Japan) under bright field illumination at 40× magnification.

### 2.10. Statistical Analysis

The statistical software used in the present study were GraphPad Prism 5.0 Software (San Diego, CA, USA) and Origin 2020b (Origin Lab—Data analysis and Graphing Software, Szeged, Hungary). The results were expressed as a mean ± standard deviation. For the MTT assays, a one-way ANOVA analysis was applied to determine the statistical differences followed by a Tukey’s multiple comparisons test (* *p* < 0.05; ** *p* < 0.01; *** *p* < 0.001).

## 3. Results

### 3.1. Physicochemical Screening of Naked MIONPs and MCS

#### 3.1.1. Structural and Nanoparticle Characterization

In order to investigate the phase composition and estimate the crystallite size, XRD was employed. Figure 1 exhibits the XRD patterns of MIONPs synthesized using the combustion method.

The peaks of MIONPs with characteristic maximum intensity were recorded at 18.97, 30.08, 35.47, 43.23, 53.69, 56.98, 62.86, 74.43, and 79.3 degrees on a 2θ axe, which indicates the magnetite main phase of the magnetic iron oxide nanoparticles. The peaks recorded at 33.13°, 49.38°, and 71.47° on the 2θ axe, indicate the formation of hematite as a minor second phase. The analyzed diffraction spectra were matched with the standard XRD peaks of Fe_3_O_4_ from JCPDS file 1,011,032 and with the standard XRD peaks of hematite with JCPDS file 9,015,964 and confirmed the formation of magnetite as the main phase.

Based on the XRD data, the structural parameters related to MIONPs (i.e., position and width of peaks, crystallite size, lattice constant, and distance between crystal planes) were calculated for the most intense peak (311) at 35.47° on 2θ, whose results are summarized in Table 1.

The Debye–Sherrer’s equation (2) measures the size of naked nanoparticles according to the broadening of the most intense peak (311–35.47°) in the XRD graph (Figure 1). Based on this equation, the diameter of the MIONP crystallites at room temperature (24 °C) were determined to be 9 nm.

#### 3.1.2. Thermal Behavior

The TG–DSC graphic of naked MIONPs is depicted in Figure 2.

Thermal analysis indicated that the total mass loss of 17.74% occurred in three stages. In the first stage, the sample lost 2.36% of total mass with very slight energy changes, almost undetectable on the DSC curve, in the range of 60–150 °C, which can be attributed to water desorption and/or dehydration of the magnetic nanopowder. The second stage revealed a loss of 14.68% of total mass with an intense exothermic effect at 347.4 °C and a lightly exothermic effect at 368.9 °C attributed to the phase transformation of Fe_3_O_4_ to γ-Fe_2_O_3_. The fade exothermic effect recorded on the DSC curve at 631.9 °C, accompanied by a slight mass loss, can be assigned to the oxidation of residual carbon present in the sample due to the combustion process.

#### 3.1.3. Magnetic Properties of Naked MIONPs and Aqueous MCS

The static magnetization properties of the naked MIONPs and aqueous MCS recorded at room temperature (24 °C) is presented in Figure 3A. The saturation magnetization of both samples was obtained at maximum field (H = 900 kA/m), manifesting a value equal to 46.26 emu/g for MIONPs and 1.24 emu/g for MCS. The MIONPs obtained via the combustion method showed superparamagnetic behavior with negligible coercivity and remanent magnetization.

The intrinsic loss power of the MCS in HF AC magnetic fields was investigated. The HF AC magnetic field frequency was 100 kHz with amplitudes of 80 Oe, 100 Oe and 120 Oe. A 1 mL sample of MCS was thermally equilibrated at 20 °C prior to each measurement and placed into an adiabatic sample holder made of polystyrene. The time dependence of the sample temperatures for 80 Oe, 100 Oe, and 120 Oe field amplitudes was measured at 300 s (Figure 3B). The MCS sample exhibited an increase of 1 °C with the increase of field amplitude, which is not a significant variation. The ILP was calculated using the standard procedure [38] using the sample’s temperature increasing rate corresponding to each magnetic field amplitude. The MCS ILP increased from 0.015 to 0.055 nH*m^2^/kg when magnetic field amplitude increased from 80 to 120 Oe (Figure 3C).

#### 3.1.4. Raman Spectroscopy Characterization

In Figure 4, the Raman spectra recorded from MIONPs in different conditions with different excitation laser lines are depicted. FT-Raman with 1064 nm excitation showed only broadened G and D bands of amorphous carbon on high backgrounds; 532 nm excitation resulted in decreased backgrounds while 632 nm excitation revealed typical iron oxide bands in the 100–800 cm^−1^ range. Details of the two key bands, components of magnetite 654 cm^−1^ and maghemite 719 cm^−1^, were inserted.

Figure 5 shows the Raman spectra recorded for MCS after drop coating deposition from the droplet edge. FT-Raman with 1064 nm excitation of bulk magnetic colloidal suspension provided only the characteristic signal of amorphous carbon superimposed with a significantly high background (not shown). Micro-Raman spectra of drop-coated depositions of the aqueous suspension were probed using multiple laser excitation lines and collecting optics. The drop coating deposition of the magnetic colloidal suspension under light microscopy revealed interesting aspects of the “coffee-ring” pattern with particular margins as highlighted in micrographs (Figure 5). Multiple spectra were collected from both the edge and interior. Increasing laser exposure time resulted in increased background only; low power and low time exposure resulted in noise only. A compromise of 10 s exposure, 10 acquisitions and a laser power of 30 mW provided an optimized Raman signal recorded in the 100 to 3200 cm^−1^ range with red laser line excitation (632.8 nm). Even so, the signal was noisy, and after background subtraction and smoothing (adjacent average 5 points) weak bands at 474, 647, 873, 992, 1247, and 1757 cm^−1^ could be observed along with the prominent bands at 1365 and 1590 cm^−1^ characteristic of amorphous carbon. The broadened weak band at 647 cm^−1^ was typical for the Fe_3_O_4_ nanoparticles, as stated above. The weak bands recorded at 873 and 992 cm^−1^ were not from iron oxide nanoparticles and could have been from organic coating (small molecules of citric acid). According to Figure 5A, it could be observed that there was also a trace around 2800 cm^−1^ that could be explained by C-H stretching vibration from citric acid molecules. It should be noted that certain band position deviations relative to the pure citric acid can be expected due to interactions of the terminal groups with the metallic nanoparticles’ surfaces.

To detect the trace amount of presumptive citric acid coating, taking into account the strong fluorescence background in the Raman spectra of samples, we further conducted a surface-enhanced Raman scattering (SERS) experiment on the MCS probe deposited on a hydrophobic slide. A small amount (50 µL) of classical citrate-reduced colloidal AgNPs prepared according to the Lee − Meisel method [39] was added to the solid MIONP powder. Subsequent SERS signals were recorded using the 532 nm laser line. A clear signature of the main bands of citric acid were detectable as shown in Figure 5. The Raman spectrum of citric acid was added for comparison (Figure 5E).

#### 3.1.5. SEM Analysis of Naked MIONPs

Figure 6 shows representative images at different orders of magnitude. According to the results obtained, the nanoparticles were found to have nanometric sizes. At a higher magnification (300 kx, Figure 6A) it can be observed that naked MIONPs have a nearly spherical shape, being uniformly distributed. From the SEM general overview micrograph (Figure 6B), one can notice the carbon content on the MIONPs’ surface. The SEM analyses were in agreement with the EDX spectrum (Figure 6C).

In order to determine the chemical composition of MIONPs, an EDX analysis was performed. Figure 6C shows the atomic percentage values for iron, oxygen, and carbon. A double concentration of oxygen can be observed, which is probably due to the copper of the oxidized grid. The microelements present in the sample were identified by the peak amplitude. The EDX spectra revealed the presence of peak amplitudes of iron in three different areas (approximately 0.7 keV, 6.4 keV, and 7.0 keV) (Figure 6C).

#### 3.1.6. TEM Analysis of MCS

Figure 7 shows the SEM and TEM images of MCS along with an EDX analysis.

The measurements of MCS were taken at higher magnifications for details/measurements). The TEM image (Figure 7B) indicates that the magnetic nanoparticles were uniformly distributed. For the specific order of magnitude (300 kx), it can be seen that the magnetic nanoparticles were monodispersed with nearly spherical shapes and an average size between 6.62 and 26.1 nm. Moreover, at 300 kx magnification, it could be observed that the magnetic nanoparticles had multifaceted surfaces with activated carbon deposits (Figure 7B). In order to determine the differences in iron, oxygen, and carbon concentrations contained in the MCS, EDX analysis was employed. The EDX analysis (Figure 7C) was in agreement with the TEM image of the MCS (Figure 7B), meaning that the colloidal suspension contains C, O, and Fe uniformly distributed.

#### 3.1.7. DLS Measurements

The dynamic light scattering (DLS) method was employed in order to assess particle size and distribution (Figure 8). The hydrodynamic diameter (D_h_) and the surface charge of the MCS were determined at three different temperature values (25 °C, 37 °C, and 43 °C). The results recorded at 25 °C regarding the intensity distribution of particles size revealed that the biocompatible magnetic colloidal suspension has a narrow size distribution with a single family of nanoparticles (nanoparticles are monomodal in nature) with a mean hydrodynamic diameter of 72.7 nm (Table 2). Moreover, zeta potentials (ζ) of the biocompatible MCS at a concentration of 52 mg/mL were also determined. The as-prepared biocompatible colloidal suspension of citric acid-based MIONPs had a higher ζ value of –45.0 (pH = 9.360 at 25 °C), which indicated a very good colloidal stability in the aqueous medium.

The aqueous MCS was measured at 37 °C and 43 °C, and it can be observed from Figure 8 and Table 2 that the narrow size distribution and monomodal nature of MCS are preserved and the mean hydrodynamic diameter is constant within PDI limits. Related to zeta potential, at 37 °C and at 43 °C the MCS showed lower values than at 25 °C, which means that the zeta potential was influenced by the increase in temperature. In conclusion, even though the zeta potential of the MCS slightly decreased with the temperature increase, its stability over time was satisfactory considering the application field of magnetic colloidal suspension. From Figure 8 it can be observed that with the temperature increase, the intensity distribution of MCS was slightly flattened and approximately symmetrical at small and large diameters. Regarding the electrophoretic mobility of suspended nanoparticles, a slight increase was observed from −3.530 × 10^−4^ at 25 °C to −4.003 × 10^−4^ at 43 °C.

### 3.2. Impact of MCS on Cell Viability under Standard and Hyperthermic Conditions

The cell viability percentage of both human adenocarcinoma cell lines (MDA-MB-231 and MCF-7) and nontumorigenic breast epithelial (MCF-10A) cells, after treatment with MCS, were evaluated under standard and hyperthermic conditions by means of MTT assays.

As depicted in Figure 9, the viability rate of both carcinoma cell lines was not inhibited after the addition of MCS at a concentration of 6 µg/mL even after hyperthermic treatment; the cells displayed a viability percentage above 95%. Increasing the concentration of MCS at 30 µg/mL and applying hyperthermic conditions induced cell viability decrement; cells manifested viability percentages of 80.235 ± 3.69% (MDA-MB-231 cells) and 89.95 ± 3.58% (MCF-7 cells).

The effect exerted by MCS on mammary epithelial cell line (MCF-10A) cells is presented in Figure 10. Cell viability appeared to not be significantly influenced by either the thermal treatment applied (the percentage of viable cells varied with only a few percentages when the temperature increased around 43 °C compared to standard conditions) or by the MCS treatment (MCF-10A cell viability was above 95% even after treatment with the highest test concentration of 30 µg/mL). Moreover, exposure of MCF-10A cells to a concentration of 6 µg/mL MCS, under both standard and hyperthermic conditions, caused a slight proliferation of the nontumorigenic MCF-10A cell population (with a cell viability rate above 105%).

### 3.3. Morphological Observations

An Olympus IX73 inverted microscope equipped with a DP74 camera (Olympus, Tokyo, Japan) was utilized to compare the microscopic aspects of the stimulated cells, with and without hyperthermia, with the typical characteristics of the control cells (untreated cells cultured under standard conditions).

#### 3.3.1. Cell Morphology Assessment with and without Hyperthermic Treatment

MDA-MB-231 cells displayed no significant morphological changes after treatment with both test concentrations (6 and 30 µg/mL) of MCS, when the standard protocols were applied (Figure 11A). However, the MCS with a concentration of 30 µg/mL induced cell shrinkage (indicated by the red arrows) under hyperthermic conditions (Figure 11B).

No morphological changes were observed in the MCF-7 cells’ aspect after treatment with MCS (concentrations of 6 and 30 µg/mL) under standard conditions (Figure 12A). Still, hyperthermia influenced the effect exerted by MCS on MCF-7 cell population where a visible cell density decrement was noticed after stimulation with the highest concentration (30 µg/mL) (Figure 12B).

The morphological aspects of the nontumorigenic breast epithelial MCF-10A cell line were not influenced by exposure to MCS at concentrations of 6 and 30 µg/mL under the temperature setups employed, which included standard conditions (Figure 13A) or hyperthermic conditions (Figure 13B), as can be easily observed by comparing the morphological features of the control cells with those of the MCS-treated cells, the MCF-10A cell line manifesting a high adherence and cobblestone morphology at high confluence.

#### 3.3.2. Cellular Detection of MCS within the Cells

Observing important morphological changes of MDA-MB-231 and MCF7 human adenocarcinoma cell lines under hyperthermic conditions, it was considered of great importance to express whether there had been a temperature-influenced accumulation of MCS inside the cells. To confirm the presence and to reveal the fate of MCS accumulation into the cells under both standard and hyperthermic conditions, Prussian blue staining was performed, which is a technique that stains iron oxide nanoparticles blue. The cells treated with MCS (concentrations of 6 and 30 µg/mL) and incubated for 24 h were compared with control cells (unstimulated cells cultured in standard conditions). MCS accumulation (blue particles) within the breast adenocarcinoma (MDA-MB-231, MCF-7) cells and nontumorigenic mammary epithelial (MCF-10A) cells could be observed, as indicated by the black arrows, in Figure 14, Figure 15 and Figure 16.

MDA-MB-231 cellular uptake presented a low rate of MCS accumulation into the cells under standard conditions for both of the test concentrations, 6 and 30 µg/mL. However, MCS accumulation within MDA-MB-231 cells was strongly enhanced under hyperthermic conditions, independent of the MCS concentration applied. Under these circumstances, the MCS was located around the nucleus, as indicated by the black arrows, but did not enter the nucleus (Figure 14).

As presented in Figure 15, the MCF-7 cell line was less responsive to MCS treatment as the internalization of MCS for both concentrations (6 and 30 µg/mL) was barely noticed, under standard conditions. Nevertheless, a slight accumulation of MCS was recorded after exposure of the test concentration of 30 µg/mL under hyperthermic conditions, which may be responsible for the low cell confluence.

As shown in Figure 16, MCS presented a considerably higher accumulation rate within the nontumorigenic breast cell line when compared to the breast adenocarcinoma cell lines (MDA-MB-231 and MCF-7) under both standard and hyperthermic conditions. However, based on the images presented in Figure 16, even though the internalization rate of MCS within the MCF-10A cells was high, especially after exposure to the test concentration of 30 µg/mL, the morphological aspects of the cells were not significantly affected.

## 4. Discussion

According to the 2011 European Commission, a structure is considered to be a nanomaterial if it is a manufactured material that contains naked, aggregated, or agglomerated particles, and its size distribution is within the 1–100 nm range. In addition, it should have one, two, or three external dimensions on the nanoscale [40]. A major technological factor of innovation regarding the manufacturing process of nanomaterials is considered to be nanotechnology, more precisely nanobiotechnology, which has brought into the spotlight new ideas with regard to the applications of nanomaterials in the medical field, such as utilizing them as therapeutic carriers for drug delivery, visualization agents in magnetic resonance imaging (MRI), or heat intermediaries in cancer thermotherapy (so-called hyperthermia) [22,41,42,43,44,45].

For intravenous or oral administrations, the Food and Drug Administration (FDA) has approved in the past few years several MIONP formulations, such as MRI visualization agents, but most of them have been taken off the market. For instance, ferumoxides (Feridex IV^®^, Endorem^TM^), which consist of a mixture of Fe_3_O_4_ and γ-Fe_2_O_3_ nanoparticles, with surfaces functionalized with dextran, were used as biological targets by the reticuloendothelial system (RES) to label liver stem cells. [46]. GastroMARK and Oral-SPION, which consist of mixtures of Fe_3_O_4_ and γ-Fe_2_O_3_ nanoparticles coated with silicon on the metallic iron surfaces, were used for gastrointestinal bowel marking but then discontinued [47]. However, currently there are some MIONPs with superparamagnetic properties that have been approved and are still on the market. For instance, NanoTherm, which features MIONPs with a magnetic core coated in aminosilanes and is employed in the hyperthermia of brain tumors, is still used in Europe [48]. Additionally, Ferucarbotran (Resovist), which possesses Fe_3_O_4_ as a core material and features surfaces coated with carboxydextran, is used as a biological target in the blood stream and has stem cell-labeling applications [49].

For magnetic targeting, magnetic nanoparticles based on iron oxides should possess some key parameters, including: (i) high surface area to maximize the drug-loading amount, (ii) nanoparticle surface functionalization with specific targeting moieties to facilitate drug distribution at the tumor site while avoiding damage to other organs, (iii) a high saturation field to provide the maximum number of signals, and (iv) synergistic activity with antitumor agents to improve the efficacy of cancer treatment by decreasing the resistance of cancerous cells [16,50,51,52]. For drug delivery, it is imperative that the magnetic core of the nanomaterial (usually magnetite, Fe_3_O_4_, or maghemite, γ-Fe_2_O_3_) to be nanocrystalline and to possess superparamagnetic behavior only in the presence of an external magnetic field, and outside of its presence the nanoparticles should no longer exhibit magnetic interactions. Nowadays, the optimization process of these MIONPs is based both on the reduction of the associated side effects (reducing the amount of systemic distribution of the drug) and also on the reduction of the required dos, by improving the drug–target precision.

Depending on the number of domains and the size distribution, there are two possible main mechanisms of MIONPs in magnetic hyperthermia when they are exposed to an alternate magnetic field (AMF): relaxation loss and hysteresis loss. In relaxation loss, single-domain superparamagnetic nanoparticles with a size of < 100 nm generate heat by Néel relaxation and Brownian relaxation [53,54,55]. In hysteresis loss, the multidomain magnetic nanoparticles follow the same direction of the external magnetic moment [56,57].

The current study presents the synthesis of magnetic iron oxide nanoparticles through the combustion method using a molar ratio of 0.6:1 oxidizing agent to fuel at 400 °C. The naked MIONPs were coated with small molecules of citric acid in order to be obtained in a stable magnetic colloidal suspension. Citric acid molecules are known to have strong binding affinity to iron oxide nanoparticles due to the carboxylate groups that interact with MIONPs.

Generally, XRD can be used to characterize the composition and crystallinity of nanoparticles as well as the average crystallite size. All the peaks of XRD patterns were analyzed and indexed using the ICDD database as a reference for comparison with magnetite, maghemite, and hematite standards. The XRD pattern of the naked MIONPs (Figure 1) showed broad diffraction peaks consistent with nano-sized crystallites, allowing for the identification of the main peaks of magnetite. The calculated crystallite size, using Scherrer equation (equation 2), of the synthesized MIONPs was 9 nm, as noted in Table 1 together with other structural parameters acquired via XRD analysis. The observed peaks at 18.97°; 30.08°; 35.47°; 43.23°; 53.69°; 56.98°; 62.86°; 74.43°, and 79.3° can be assigned to (111), (220), (311), (400), (422), (511), (440), (533), and (444) planes of magnetite nanoparticles in cubic phases [19,58]. The peaks observed at 33.13°; 49.38°, and 71.47° can be assigned to (104), (024), and (1010) diffraction planes of hematite nanoparticles, which means that in the MIONPs sample we had a mixture of magnetite and hematite nanoparticles. This is not disturbing because the hematite nanoparticles do not show magnetic moment, therefore, by magnetic decantation, the hematite nanoparticles remain in suspension and the neodymium magnet does not attract them. Hence, during the nanoparticle washing process, the hematite nanoparticles would be removed. Similar results were published by Cursaru and coworkers [59].

Some of these peaks matched closely with the XRD pattern of γ-Fe_2_O_3_, but the black color of MIONPs and the peaks recorded at 74.43° and 79.3° on the 2θ scale confirmed the presence of magnetite as the main phase. Our results regarding the diameter of the MIONP crystallites determined at room temperature (24 °C) and the peaks observed on the 2θ scale are in agreement with literature data [60] and confirmed that combustion synthesis of MIONPs leads to magnetite formation as the main phase having small crystallite size.

Regarding the TG–DSC curves, they indicated various mass loss in different temperatures regions accompanied by endothermic or exothermic effects. Generally, the endothermic processes are attributed to phase transition or phase reduction but mostly to decomposition reactions, whereas exothermic processes are highlighted due to the oxidation and/or decomposition reactions and crystallization reactions. In the present study, the TG–DSC spectra of naked MIONPs exhibited two exothermic processes at 347.4 °C and at 368.9 °C attributed to the phase transformation of Fe_3_O_4_ to γ-Fe_2_O_3_. Another slightly exothermic process is localized at 631.9 °C and can be associated with the oxidation of residual carbon present in the magnetic sample. A similar effect of magnetite oxidation was reported in the literature, the process being induced by laser irradiation [61].

In order to provide a structural fingerprint by which the molecules contained in MIONPs and MCS via their atomic vibrations can be identified, Raman spectroscopy was employed. Raman spectra were challenging to record due to the inherently low scattering of predominantly ionic interactions within MIONPs, regardless of excitation line and acquisition parameters. As observed from the thermogravimetry results, magnetite in phase easily transforms into maghemite upon heat input—the same occurs under focused laser energy. Nevertheless, the dominant Raman peaks were observed in the range of 100–800 cm^−1^, more precisely at 654 cm^−1^ for Fe_3_O_4_ and at 719 cm^−1^ for γ-Fe_2_O_3_. Our structural and methodological observations were in agreement with literature data [62]. The Raman spectrum of MIONPs showed broadened G and D bands, which are characteristic to amorphous carbon (Figure 4), besides the typical iron oxide bands in the 100–800 cm^−1^ range, probably resulting from the combustion reaction residue that was not washed, consistent with SEM observations of carbon deposits. Nevertheless, EDX showed that the weight content of amorphous carbon was negligible compared to iron oxide comprising Fe and O. The coating of citric acid onto MIONPs was also confirmed using Raman spectroscopy, as can be seen in Figure 5, by the weak bands recorded at 873 and 992 cm^−1^ assigned to organic coating, and also by the trace around 2800 cm^−1^. which could be due to C-H stretching vibration from citric acid molecules (Figure 5A).

The solid magnetic nanoparticles and the aqueous suspension were characterized by employing consecrated analytical techniques such as SEM and TEM imaging and DLS measurements, respectively. The SEM imaging (Figure 6A) of MIONPs showed that the magnetic nanoparticles had polygonal shapes distributed uniformly. In addition, the carbon deposits on MIONP surfaces were also highlighted (Figure 6B). The carbon impurities were most likely the result of incomplete fuel oxidation [63]. The values recorded for the size of the MCS (Figure 7B) were in the range of 6.62–26.1 nm, which could indicate the presence of high polydispersity that would be in accordance with the data obtained from DLS measurements (Table 2).

Due to the fact that superparamagnetic behavior is an important parameter for biomedical applications, the saturation magnetization of MIONPs and MCS were evaluated. A higher decrease of saturation magnetization in the case of MIONPs (46.26 emu/g) compared to MCS (1.24 emu/g) was recorded. In the last decade, solid magnetic nanoparticles or colloidal suspensions have been widely used as heating intermediates in the treatment of cancer due to their ability to release heat under an AC magnetic field at a specific amplitude and frequency. Figure 3B shows the temperature changes as a function of time for the MCS sample. In order to have a better understanding of the hyperthermic behavior of MCS, the heating capacity of MCS was investigated via ILP analysis. The heating capacity revealed a value of 0.055 nH*m^2^/kg at a magnetic field amplitude of 120 Oe (Figure 3C), which increased along with increases in the magnetic amplitude field. This value indicates that the obtained MCS could be used in magnetic heat generation.

Size and zeta potential measurements of MCS at different temperatures are summarized in Table 2 (n = 5 for the sample). According to Table 2, the z-average was almost constant when the temperature increased while zeta potential decreased when the temperature increased. The average hydrodynamic diameter (72.7 nm at 25 °C) was bigger than the core size of CA-MIONPs (26.1 nm) observed in the TEM image in Figure 7B. According to literature data, colloidal suspensions with ζ values below −30 mV or above +30 mV can lead to strong electrostatic repulsion between nanoparticles, ensuring system stability. A ζ value near 0 mV leads to magnetic nanoparticle flocculation followed by sedimentation [34]. Moreover, several studies have demonstrated that MIONPs with positively charged surfaces are considered as having a higher toxicity in the biomedical field [4,64,65]. Based on the presence of tri-carboxyl (–COOH) groups with negative charge on the surfaces of the nanoparticles, citric acid acted as stabilizer, thus preventing the aggregation of MIONPs. Considering this, it was expected that using citrate molecules on the nanoparticles’ surfaces would induce electrostatic repulsion between MIONPs, thus leading to an increased dispersion of nanoparticles in the reaction media and also generating good stabilization in the aqueous media. The obtained MCS was highly negatively charged (−38.7 mV at 43 °C) (Table 2). Moreover, the high zeta potential value confirmed the presence of citrate ions on the surface of MIONPs assuring long-term stability of the MCS. The value regarding zeta potential obtained by our group was higher than in most of the data published in the specialized literature regarding the synthesis of MIONPs and surface functionalization with citric acid. For instance, Nigam and coworkers [66] prepared citric acid functionalized Fe_3_O_4_ nanoparticles using the soft chemical route, obtaining an aqueous colloidal suspension based on magnetic nanoparticles with a hydrodynamic diameter of 25 nm whose zeta potential increased as the pH increased (−25.5 mV at pH 6). Additionally, De Sousa and coworkers [67] synthesized magnetite nanoparticles using a chemical route electrostatically stabilized by citric acid coating with hydrodynamic sizes in the range 17–30 nm and dispersed in an aqueous solution. The authors found that a pH > 4 resulted in the largest mean value of ζ, close to −36 mV. The electrophoretic mobility of suspended nanoparticles is determined by electrophoretic light scattering (based on the principle of electrophoresis). By applying an electric field to electrically charged nanoparticles in suspension, they were able to change to the opposite sign electrode at a certain speeds. In order to determine the speed of the nanoparticles, or their mobility, they were irradiated by a laser, after which the Doppler effect of the light scattered by the moving nanoparticles was observed. Regarding the mobility of the suspended nanoparticles, this was dependent on the viscosity of the carrier liquid (distilled water), on the size of the suspended nanoparticles, and their charge. In the present study, the mobility increased as the temperature increased (Table 2).

The study performed by Mai and coworkers [19] is in good agreement with the current study. Mai’s group synthesized magnetic iron oxide nanoparticles using the coprecipitation method. After the surfaces of MIONPs were functionalized by coating them with small molecules of citric acid, their cytotoxicity was determined via reactive oxygen species (ROS) generation. The hydrodynamic diameter of CA-MIONPs obtained by coprecipitation was 87 nm with a polydispersity index of 0.225 and zeta potential of −35.2 mV vs. the MCS obtained in our study using the combustion method: 72.7 nm with a PDI of 0.179 and a zeta potential of −45.0 mV, all measured at 25 °C. Since the PDI can vary from 0.01 up to 0.5 [68], it can be concluded that the nanoparticles contained in aqueous MCS obtained in this study were reasonably monodispersed, a fact that is highlighted by the narrow distribution of MCS (Figure 8) together with its TEM imaging (Figure 7B).

Since the main objective for the development of MCS was its addressability for bio-medical applications, especially as tools for inducing hyperthermic conditions in the biological environment, an evaluation of the MCS biological profile by employing an in vitro model under both normothermic (37 °C) and hyperthermic (43 °C) conditions was a mandatory aspect of the present study. The in vitro model consisted of two genotypically different human breast adenocarcinoma cell lines: one highly aggressive and nonhormone dependent (estrogen and progesterone negative) cell line, MDA-MB-231, and one hormone responsive (estrogen and progesterone positive) cell line, MCF7. In addition, one nontumorigenic human breast epithelial cell line (MCF-10A) was used to obtain a more complex and insightful biological profile.

Thus, the biological data obtained through the colorimetric MTT assays revealed that MCS at a concentration of 30 µg/mL induced significant cell viability reduction of the highly aggressive breast adenocarcinoma MDA-MB-231 cells under hyperthermic conditions, whereas the hormone responsive breast adenocarcinoma MCF7 cells and the nontumorigenic MCF-10A cell line manifested a good cell viability rate (Figure 9 and Figure 10). The distinctive responses expressed by the breast adenocarcinoma cell lines after exposure to MCS could be caused by the different metabolic activity rates of the cells [69], as it has been observed that the highly aggressive breast adenocarcinoma cell line presents a higher sensibility to magnetite exposure and a higher uptake rate compared to MCF7 cells that manifest a good resistance to these types of nanoparticles, an aspect that may be related to the 50% lower accumulation rate of iron within MCF7 cells [70].

These aspects were also corroborative with the data obtained through Prussian blue staining (Figure 14 and Figure 15) where the accumulation rate of MCS could be easily ob-served. Moreover, the results confirmed the higher uptake rate of Fe_3_O_4_ nanoparticles by MDA-MB-231 cells compared to MCF7 cells, a phenomenon that was potentiated under hyperthermic treatment and can be correlated with the cell viability decrease observed in MDA-MB-231 cells.

Our research group has already demonstrated that Fe_3_O_4_ nanoparticles are responsible for inducing alterations of amelanotic human A375 cells and murine melanoma B164A5 cells depending on cell phenotype and metastatic activity [71,72,73]. However, through the current study, antitumorigenic screening of these nanoparticles extends to breast adenocarcinoma cells, and based on the present data, MCS developed in this study revealed promising potential as a hyperthermia-inducing tool in the biological environment. Moreover, it developed good activity as an antitumor strategy under hyperthermic conditions, especially in the highly aggressive human breast adenocarcinoma MDA-MB-231 cells. Nevertheless, for an enhanced antitumor effect, MCS as developed herein could be employed as hyperthermia tools in various magnetically responsive formulations containing antitumor agents that target breast cancer therapy, such as nanohydrogels [74], solid lipid nanoparticles [75], magnetoliposomes [76,77], or other smart carriers for the combined approach of hyperthermia and chemotherapy.

## 5. Conclusions

Superparamagnetic iron oxide nanoparticles were prepared via the combustion route using citric acid as fuel. These nanoparticles were stabilized with citric acid, leading to a stable monodispersed aqueous colloidal suspension with a mean hydrodynamic diameter of 72.7 nm, a PDI of 0.179, and a zeta potential of −45.0 mV. The magnetic colloidal suspension can be used in magnetic heat generation, as evidence by the ILP analysis. The heating capacity revealed a value of 0.055 nH*m^2^/kg at the highest magnetic field amplitude applied and increased along with increases in the amplitude field. Regarding the biological profile of the MCS, the viability of the nontumorigenic MCF-10 cell line was not affected after exposure to MCS at a concentration of 30 µg/mL, whereas the highly aggressive breast adenocarcinoma MDA-MB-231 cell line manifested statistically significant cell viability reduction under hyperthermic conditions, thus endorsing the usefulness of the MCS as a hyperthermic agent addressing antitumor therapy.

## Figures and Tables

**Figure 1 nanomaterials-11-01189-f001:**
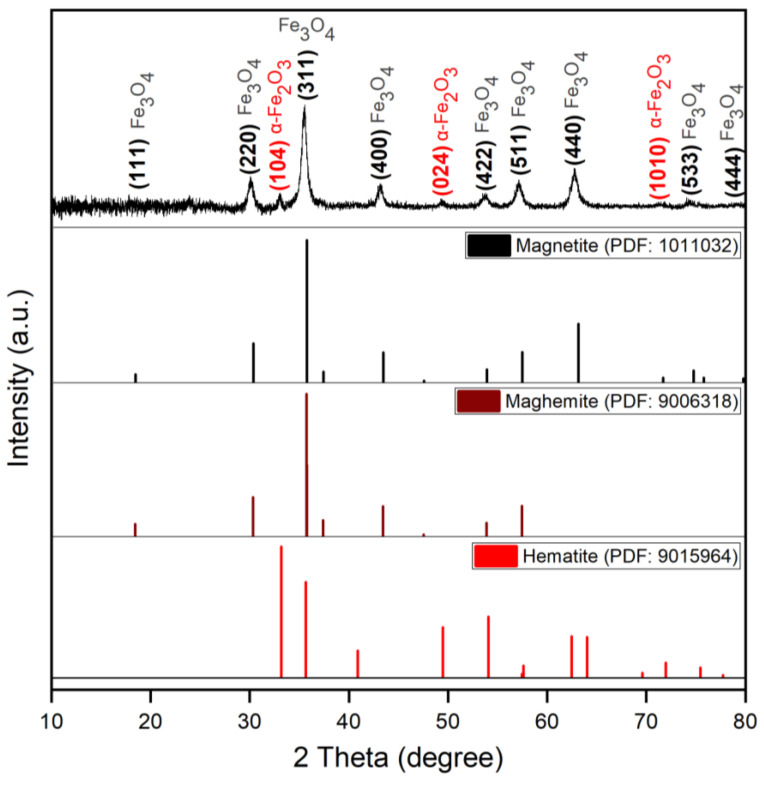
XRD pattern of naked MIONPs compared to the XRD patterns of magnetite (black), maghemite (brick), and hematite (red) from the International Centre for Diffraction Data Powder Diffraction File (ICDD PDF) 4+ 2019 data.

**Figure 2 nanomaterials-11-01189-f002:**
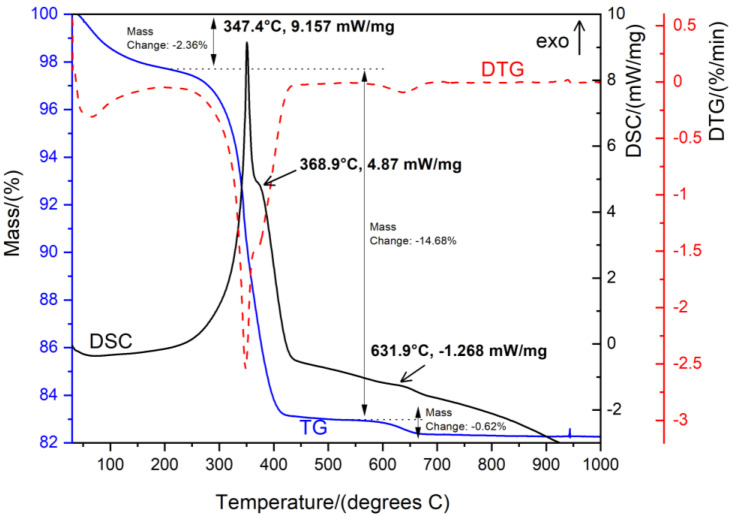
TG–DSC curves of naked MIONPs.

**Figure 3 nanomaterials-11-01189-f003:**
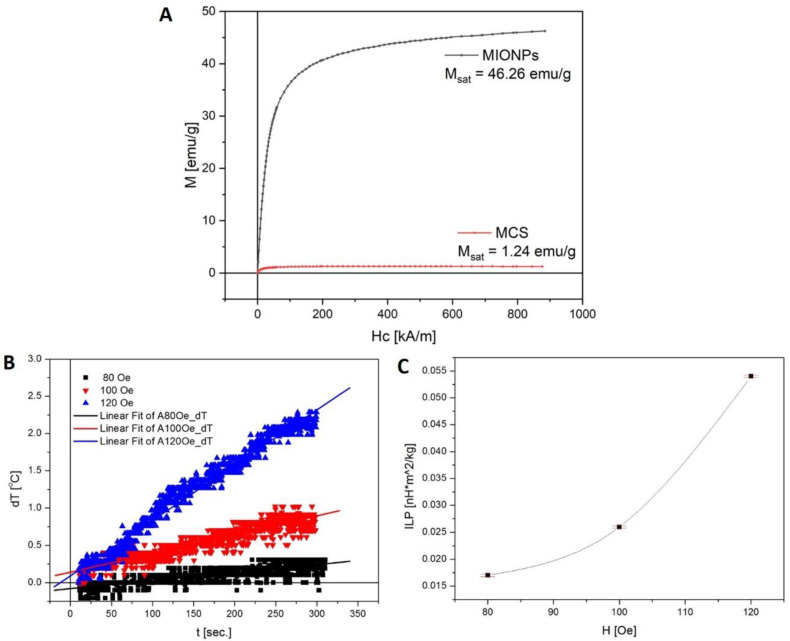
(**A**) Magnetization curves of MIONPs and MCS, (**B**) Time dependence of MCS temperature, increased for 80, 100, and 120 Oe HF AC magnetic field amplitudes, and (**C**) HF AC magnetic field amplitude dependence of MCS sample ILP.

**Figure 4 nanomaterials-11-01189-f004:**
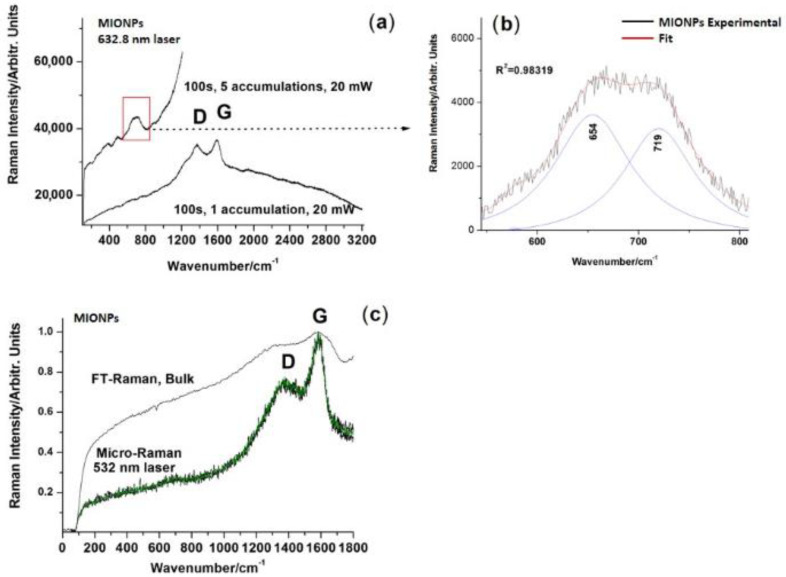
Typical Raman signals collected from the MIONPs in different conditions with different excitation laser lines: (**a**) under 632.8 nm excitation, (**b**) zoom-in of the 550–800 cm^−1^ region showing the deconvolution of the spinel phase marker band, revealing modes of magnetite at 654 and maghemite at 719 cm^−1^, (**c**) FT-Raman under 1064 nm and micro-Raman under 532 nm excitation.

**Figure 5 nanomaterials-11-01189-f005:**
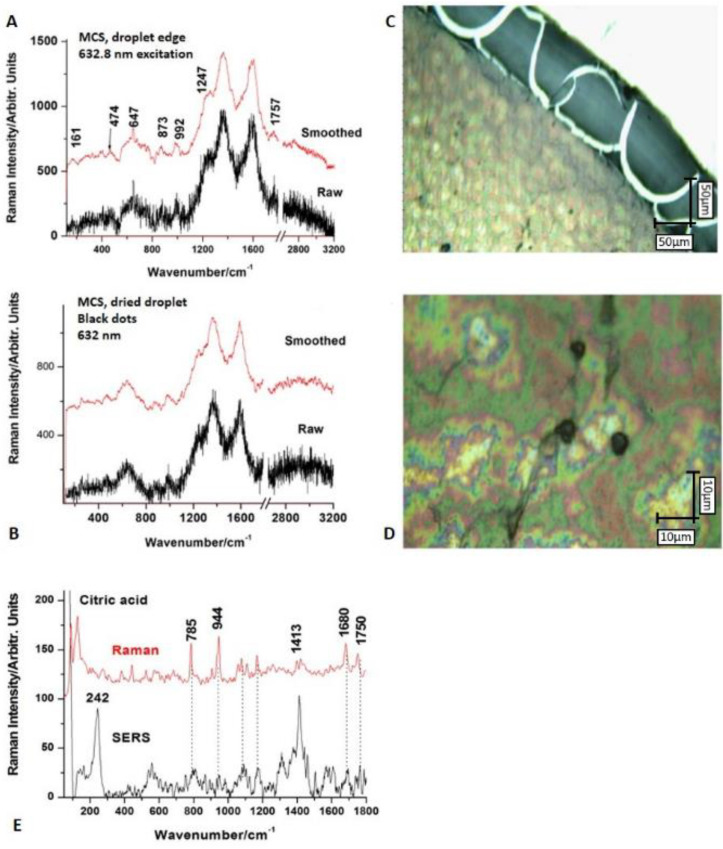
Typical Raman spectra collected from MCS (**A**) and black MIONP aggregates from the droplet (**B**) the light deposits also revealed amorphous carbon signals. Excitation: 632 nm. The droplet edge and center under light microscopy are highlighted in micrographs (**C**,**D**), respectively. Scale bar 50 µm (**C**) and 10 µm (**D**). (**E**) SERS signal recorded from a drop-coated deposition of MCS after adding the plasmonic silver nanoparticles (AgNPs) to quench the fluorescence: clear bands characteristic of citric acid were visible; Raman spectrum of citric acid is given for comparison.

**Figure 6 nanomaterials-11-01189-f006:**
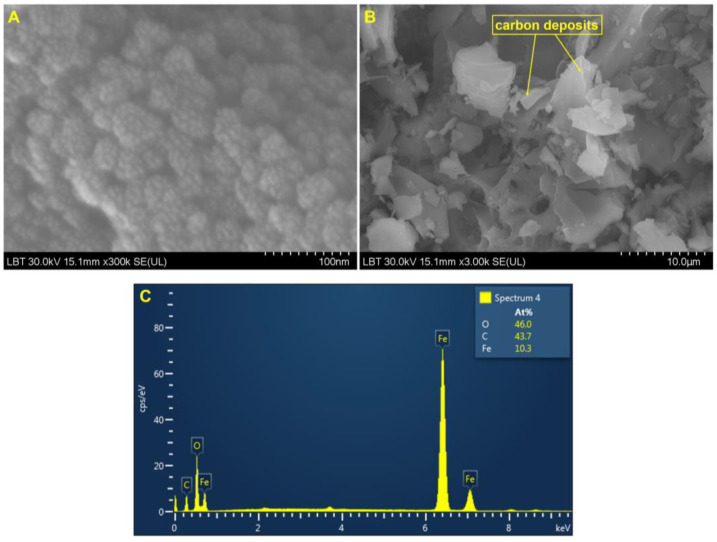
SEM images for MIONPs synthesized via the combustion method at different orders of magnitude: scale bars (**A**) 100 nm, (**B**) 10 µm, and (**C**) EDX spectrum.

**Figure 7 nanomaterials-11-01189-f007:**
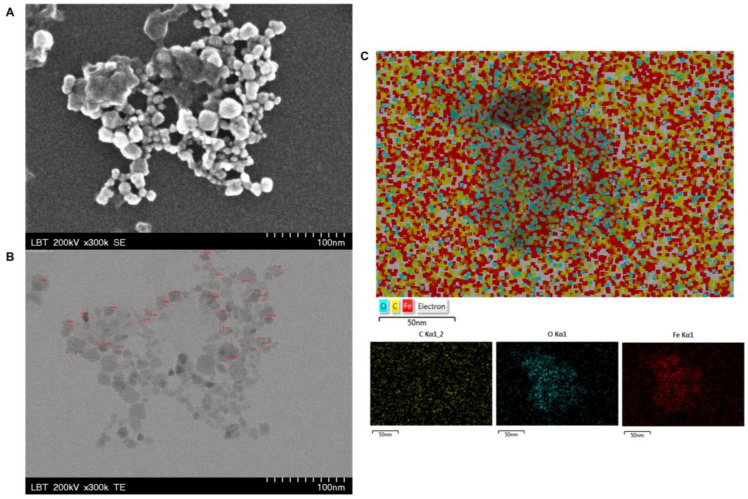
SEM (**A**) and TEM (**B**) images at higher magnifications for details/measurements, and an EDX map (**C**) analysis of MCS.

**Figure 8 nanomaterials-11-01189-f008:**
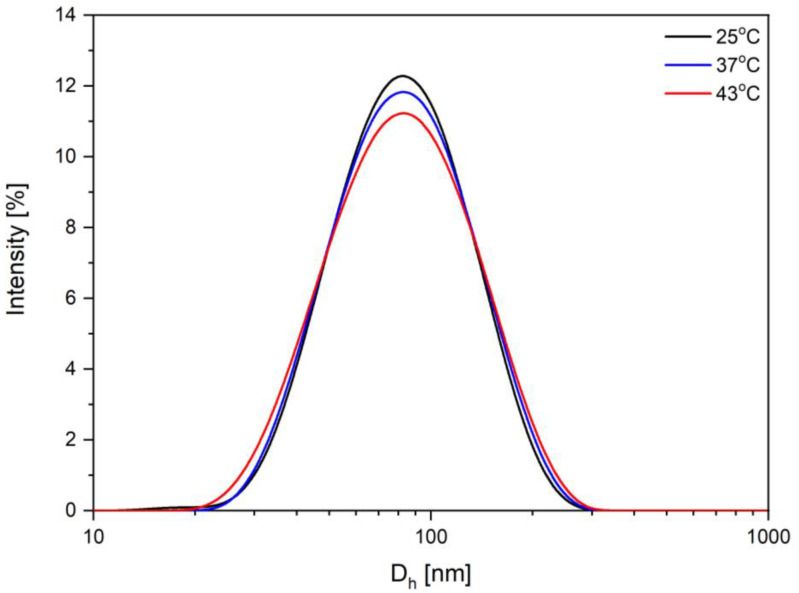
The intensity distribution of particle size contained in aqueous MCS.

**Figure 9 nanomaterials-11-01189-f009:**
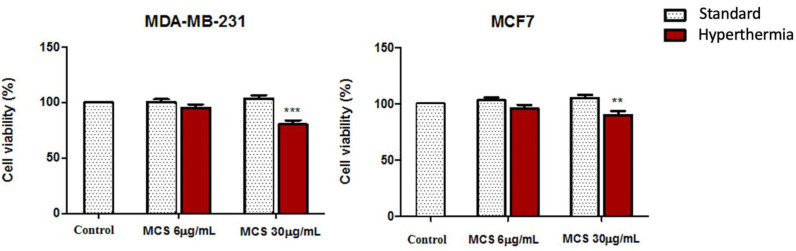
Cell viability assessment of breast adenocarcinoma cell lines (MDA-MB-231 and MCF-7) after treatment with MCS at concentrations of 6 and 30 µg/mL, 24 h post-stimulation. The experiments were performed under standard and hyperthermic conditions. The cell viability percentage, with and without hyperthermia, was normalized to control cells (no stimulation and incubated in standard conditions). The data represent the mean values ± SD of three independent experiments. One-way ANOVA analysis was applied to determine the statistical differences followed by Tukey’s multiple comparisons test (** *p* < 0.01; *** *p* < 0.001 *versus* control cells).

**Figure 10 nanomaterials-11-01189-f010:**
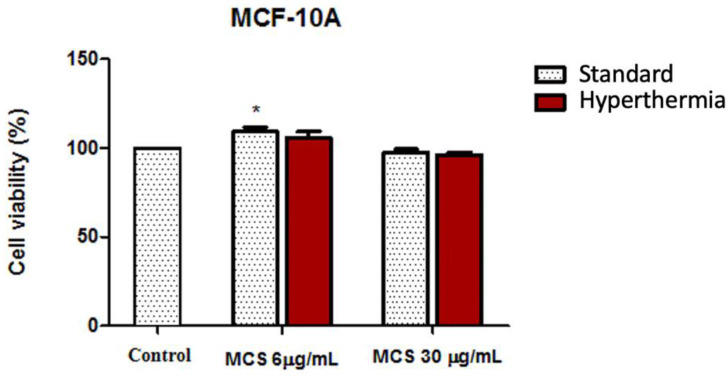
Cell viability assessment of nontumorigenic breast epithelial cell line (MCF-10A) after treatment with MCS at concentrations of 6 and 30 µg/mL, 24 h post-stimulation. The experiments were performed under standard and hyperthermic conditions. The cell viability percentage, with and without hyperthermia, was normalized to control cells (no stimulation and incubated in standard conditions). The data represent the mean values ± SD of three independent experiments. One-way ANOVA analysis was applied to determine the statistical differences followed by Tukey’s multiple comparisons test (* *p* < 0.05 versus control cells).

**Figure 11 nanomaterials-11-01189-f011:**
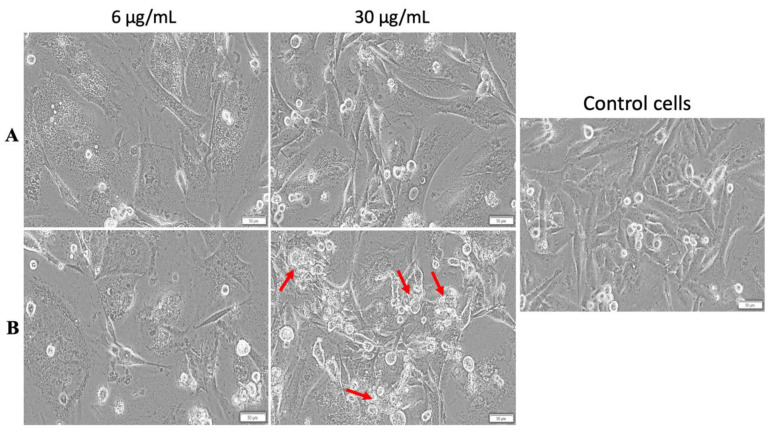
Morphology of MDA-MB-231 cells after treatment with MCS under standard (**A**) and hyperthermic conditions (**B**). Phase contrast pictures were taken 24 h postincubation at a magnification of 20×. Scale bars represent 50 µm. Cells undergoing morphological changes are marked with red arrows.

**Figure 12 nanomaterials-11-01189-f012:**
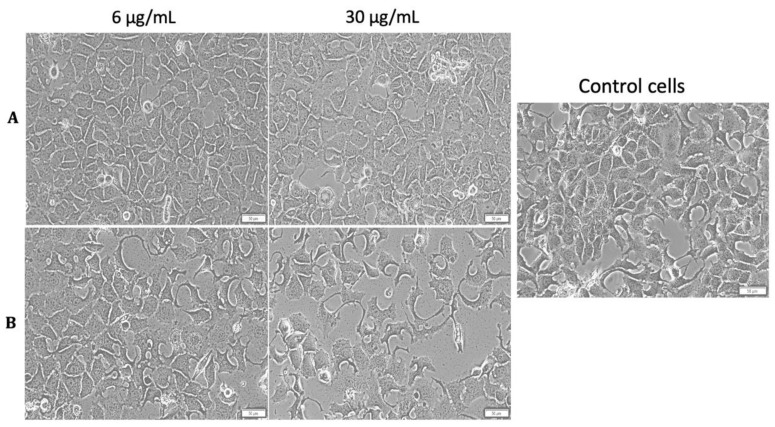
Morphology of MCF-7 cells after treatment with MCS under standard (**A**) and hyperthermic conditions (**B**). Phase contrast pictures were taken 24 h post-incubation at a magnification of 20×. Scale bars represent 50 µm.

**Figure 13 nanomaterials-11-01189-f013:**
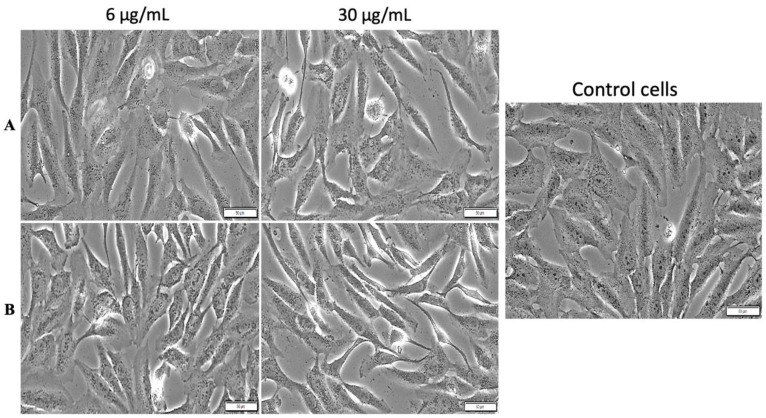
Morphology of MCF-10A cells after treatment with MCS under standard (**A**) and hyperthermic conditions (**B**). Phase contrast pictures were taken 24 h post-incubation, at a magnification of 20×. Scale bars represent 50 µm.

**Figure 14 nanomaterials-11-01189-f014:**
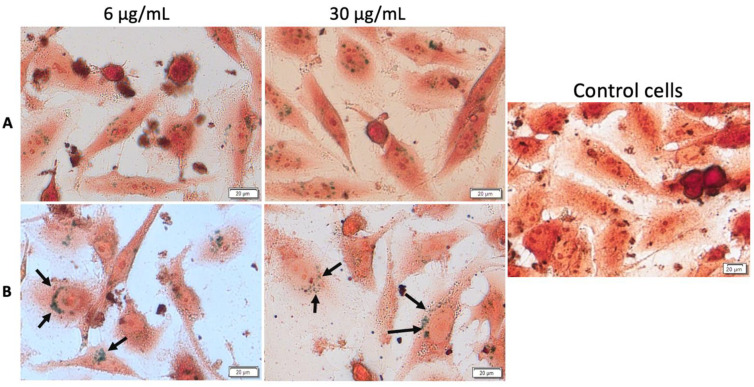
Cellular internalization of MCS inside MDA-MB-231 cells under standard (**A**) and hyperthermic conditions (**B**). The cells were visualized under bright field microscopy at a magnification of 40×. Scale bars represent 20 µm.

**Figure 15 nanomaterials-11-01189-f015:**
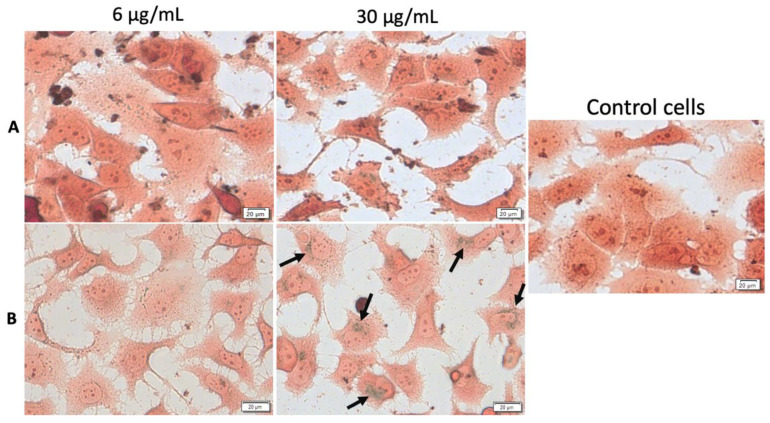
Cellular internalization of MCS within MCF-7 cells under standard (**A**) and hyperthermic conditions (**B**). The cells were visualized under bright field microscopy at a magnification of 40×. Scale bars represent 20 µm.

**Figure 16 nanomaterials-11-01189-f016:**
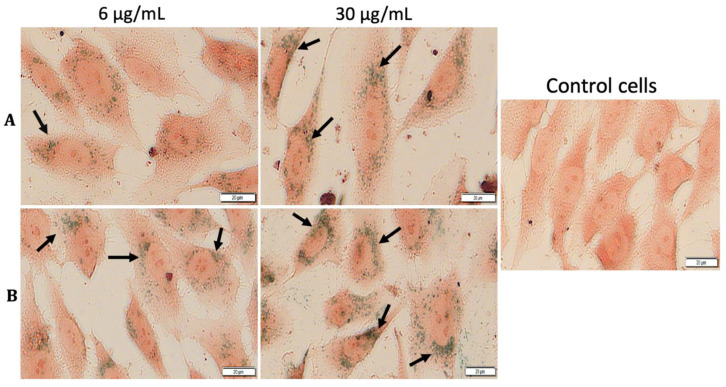
Cellular internalization of MCS within MCF-10A cells under standard (**A**) and hyperthermic conditions (**B**). The cells were visualized under bright field microscopy at a magnification of 40×. Scale bars represent 20 µm.

**Table 1 nanomaterials-11-01189-t001:** Structural parameters of XRD analysis for the most intense peak (311).

	2θ [deg.]	FWHM [deg.]	D_XRD_ [nm]	d_hkl_ [Å]	a [Å]
MIONPs	35.47	0.584	9	2.5285	8.3200

**Table 2 nanomaterials-11-01189-t002:** Structural characterization parameters of MCS based on MIONPs dispersed in distilled water recorded at different temperatures.

Sample/Carrier Liquid/Temperature	Z-Ave Diameter (nm) (Cumulants Results)	PDI (Polydispersity Index)	Zeta Potential (mV)	Mobility (cm^2^/Vs)
MCS/water/25 °C	72.7	0.179	−45.0	−3.530 × 10^−4^
MCS/water/37 °C	73.5	0.176	−41.3	−3.963 × 10^−4^
MCS/water/43 °C	72.8	0.183	−38.7	−4.003 × 10^−4^

## Data Availability

Not applicable.

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
