# Peer review of "Biocompatible Magnetic Colloidal Suspension Used as a Tool for Localized Hyperthermia in Human Breast Adenocarcinoma Cells: Physicochemical Analysis and Complex In Vitro Biological Profile"

_nanomaterials, 2021, doi:10.3390/nano11051189_

Round 1
Reviewer 1 Report
The manuscript nanomaterials-1207810 devoted the actual field of the medical science, namely study of biocompatible magnetic colloidal suspension as a tool for human breast adenocarcinoma traetment and can be interested to the specialists working in this field. The author’s opinion is clear. I am personally impressed by the structure of the article, the systematization of scientific data and the sequence of its presentation. Work is performed at sufficient scientific level and has good quality; the results of investigation are professionally interpreted. The paper fit the Journal scope and formal requirements. My decision is accept
Author Response
The authors of this manuscript would like to express their gratitude towards the reviewer that evaluated so highly the paper.
Reviewer 2 Report
Report of Manuscript nanomaterials-1207810 for Nanomaterials
Title: Biocompatible magnetic colloidal suspension used as a tool for localized hyperthermia in human breast adenocarcinoma cells: Physicochemical analysis and complex in vitro biological profile by Elena-Alina Moacăet al.
The present work reports a facile single-step process for preparation of highly stable and biocompatible magnetic colloidal suspension based on citric acid coated magnetic iron oxide nanoparticles, used as effective heating source for the hyperthermia treatment of cancer cells.
The paper is well-written, the English is good and I think that this work could be of interest for the field of basic and applicative research of NPs. The paper is technically sound and it is of great interest. The work is well structured and the proposed goals were achieved. I really appreciated the combination of "characterization" and "applicability".
The manuscript contains new information to justify publication. The methods described comprehensively. The list of references should be improved and modified. The interpretations and conclusions justified by the results. Different and complementary analyzes were carried out. The manuscript appears complete in all its parts.
However, the manuscript should be improved and it will be worth for publication after some “minor” revisions as recommended below.
Abstract - The abstract is too long. I ask the authors to rewrite it. Focus on goals and motivations. Furthermore, I suggest that authors avoid acronyms in the abstract.
Keywords – Select only 4/5 keywords.
Lines 38-41 – About magnetic nanoparticles, I suggest increasing these references. The authors show several applications of nanoparticles. I suggest reverencing recent literature also within “MDPI Nanomaterials”. This research field is very vast and of great scientific interest. I suggest for example to add “https://doi.org/10.3390/nano10101919”; “https://doi.org/10.3390/nano11030627”; https://doi.org/10.3390/nano10112310 and others.
Sections 2.2 and 2.3 – The authors should indicate the purity of the reagents used.
In the whole text
-Please, define the room temperature.
-I ask the authors to review the experimental errors. Often the reported measurements appear to be error-free. Check the entire manuscript carefully.
-I ask the authors to consider adding more biological details. Some notions are unclear to non-biologically educated readers.
Conclusions - The conclusions need to be enriched to emphasize the applicability of the results found: this aspect is fundamental to the publication and impact of this manuscript.
-Different minor typo-corrections that should be performed.
This reviewer hopes to receive a new and improved version of the manuscript. The results and details must be particularly emphasized.
Author Response
Response: Firstly, we would like to thank the reviewer for evaluating our manuscript. We appreciate all the pertinent remarks made by the reviewer and we know that by responding to these queries we will obtain an improved manuscript.
Q1.Abstract - The abstract is too long. I ask the authors to rewrite it. Focus on goals and motivations. Furthermore, I suggest that authors avoid acronyms in the abstract.
R1. We modified the abstract entirely, the abstract has now 193 words (before were 230 words), we removed the acronyms and underlined the goals and motivations as the reviewer recommended. In the manuscript, the abstract now in marked with red color, Line 35-49.
Q2.Keywords – Select only 4/5 keywords.
R2. We adjusted the number of keywords to five, and they are: magnetic iron oxides nanoparticles, citric acid, Raman spectroscopy, combustion method, breast adenocarcinoma; they are marked with red in the manuscript, Line 50-51.
Q3.Lines 38-41 – About magnetic nanoparticles, I suggest increasing these references. The authors show several applications of nanoparticles. I suggest reverencing recent literature also within “MDPI Nanomaterials”. This research field is very vast and of great scientific interest. I suggest for example to add“http://dx.doi.org/10.3390/nano10101919”; “http://dx.doi.org/10.3390/nano11030627”; http://dx.doi.org/10.3390/nano10112310 and others.
R3. We added two new paragraphs in the manuscript- L103-118 and L125-132, in which we introduced the suggested references and other that we considered appropriate. The paragraphs are also marked in red in the manuscript.
Q4.Sections 2.2 and 2.3 – The authors should indicate the purity of the reagents used.
R4. We added in text the purity for iron oxide nonahydrate ≥ 96% ad for the citric acid monohydrate ≥ 99.5% purity. Can be seen at L153 and L154, marked in red.
Q5.In the whole text
-Please, define the room temperature.
-I ask the authors to review the experimental errors. Often the reported measurements appear to be error-free. Check the entire manuscript carefully.
-I ask the authors to consider adding more biological details. Some notions are unclear to non-biologically educated readers.
R5. The value of the room temperature was added in the whole text as 24 °C.
All manuscript was checked for errors and correction were applied. Minor modifications can be noticed at the followings lines: L55, L56, L68, L64, L74, L85-86, L92, L101, L122-123, L137, L153, L154, L157, L159, L160, L168, L169, L200-203, L207-209, L216, L227, L258, L260, L268, L299, L305, L307, L308, L318, L321, L331, L350, L352, L353, L356, L358, L359, L362, L365-367, L381, L385, L451, L471, L475, L508, L592-593, L603, L605, L684, L694-698, L700-701, L708, L712, L713, L715, L723, L754, L769-771, L802, L852.. All the changes are made in red.
Details regarding the in vitro biological experiment were inserted at the experimental part as new subchapters namely 2.6 and 2.7 between the lines 280-298. In addition, for a better understanding of the biological details was added the formula employed for quantification of cell viability rate between the lines L311-L317.
Q6.Conclusions - The conclusions need to be enriched to emphasize the applicability of the results found: this aspect is fundamental to the publication and impact of this manuscript.
R6. At the reviewer suggestion, the conclusions were rewritten, L854-859.
Q7.-Different minor typo-corrections that should be performed.
R7. The entire manuscript was checked for typos and corrections were applied.
This reviewer hopes to receive a new and improved version of the manuscript. The results and details must be particularly emphasized.
We hope that the new and improved version of the manuscript would please the reviewer.